# Pro-atrial natriuretic peptide and pro-adrenomedullin before cardiac surgery in children. Can we predict the future?

Sara Bobillo-Perez[1,2]☺, Monica Girona-Alarcon[1,2]☺, Patricia Corniero[1‡], Anna Sole-Ribalta[1,2‡], Monica Balaguer[1,2‡], Elisabeth Esteban[1,2‡], Anna Valls[3‡], Iolanda Jordan [1,4]☺*, Francisco Jose Cambra[1,2‡]

**1** Pediatric Intensive Care Unit, Hospital Sant Joan de Déu, Universitat de Barcelona, Barcelona, Spain, **2** Disorders of Immunity and Respiration of the Pediatric Critical Patient Research Group, Institut de Recerca Hospital Sant Joan de Déu, Universitat de Barcelona, Barcelona, Spain, **3** Biochemistry Laboratory, Hospital Sant Joan de Déu, Universitat de Barcelona, Barcelona, Spain, **4** Pediatric Infectious Diseases Research Group, Institut de Recerca Hospital Sant Joan de Déu, CIBERESP, Barcelona, Spain

☺ These authors contributed equally to this work.
‡ These authors also contributed equally to this work.
* ijordan@sjdhospitalbarcelona.org

## Abstract

### Introduction and objective

Pro-atrial natriuretic peptide (proANP) and pro-adrenomedullin (proADM) levels increase in acute heart failure and sepsis. After cardiac surgery, children may require increased support in the intensive care unit and may develop complications. The aim of this study was to evaluate the utility of proANP and proADM values, determined prior to cardiac surgery, for predicting the need for increased respiratory or inotropic support during the post-operative period.

### Methods

This was a prospective study in children. Biomarkers were analyzed before surgery using a single blood test. The primary endpoints were the need for greater respiratory and/or inotropic support during the post-operative period. Secondary endpoints were the relationship between these biomarkers and complications after surgery.

### Results

One hundred thirteen patients were included. ProANP and proADM were higher in children who required greater respiratory and inotropic support, especially proANP; for increased respiratory support, 578.9 vs. 106.6 pmol/L (p = 0.004), and for increased inotropic support, 1938 vs. 110.4 pmol/L (p = 0.002). ProANP had a greater AUC than proADM for predicting increased respiratory support after surgery: 0.791 vs. 0.724. A possible cut-off point for proANP could be ≥ 325 pmol/L (sensitivity = 66.7% and specificity = 88.8%). In the multivariate analysis, the logarithmic transformation of proANP was independently associated with the need for increased respiratory support (OR = 3.575). Patients who presented a poor

**Data Availability Statement:** All relevant data are within the manuscript and its Supporting Information files.

**Funding:** The funders had no role in study design, data collection and analysis, decision to publish, or preparation of the manuscript.

**Competing interests:** The authors have declared that no competing interests exist.

outcome after cardiac surgery also had higher biomarker values (proADM, p = 0.013; proANP, p = 0.001).

## Conclusions

Elevated proANP before cardiac surgery may identify which children will need more respiratory and inotropic support during the post-operative period.

## Introduction

Cardiac surgeries in childhood are complex procedures in which every detail is important. On the one hand, the result of an intervention will vary depending on the type of congenital heart defect (CHD), the patient's situation before cardiac surgery, and the characteristics of the surgical techniques. On the other hand, optimal management after the surgery in the Intensive Care Unit (ICU) is also essential. In the ICU, multi-parameter monitoring, together with physical examinations, provide fundamental information about the situation of the patient, but they are limited in that they only tell us what is happening at that moment. Thus, any pre-operative information that could help us predict the evolution of the patient after cardiac surgery would be extremely valuable. One example of this is the widespread use of pre-surgery scores for predicting mortality risk after heart surgery; these are useful tools that allow for risk stratification prior to the intervention [1]. During the pre-surgical evaluation, the physical exploration and the echocardiogram can also help to estimate the risk of each patient by looking for signs of volume overload or heart failure.

Another great addition to our toolbox would be analyzing specific biomarkers before cardiac surgery, thus yielding useful and exact information. One such biomarker is adrenomedullin, a vasodilator protein with multiple functions, including the regulation of pulmonary blood flow. It has been associated with poor prognosis in children and adults with septic shock, and also in patients with cardiac failure [2–5], because of its functions as a compensatory protein: it stimulates myocardial contractility and coronary blood flow, increasing cardiac output. Mid-regional pro-adrenomedullin (proADM) levels rise in step with those of adrenomedullin, but it is a more stable peptide and is also easier to measure in the laboratory, which entails a great clinical advantage [6]. The other biomarker that could be of interest here is atrial natriuretic peptide, detected in the laboratory as mid-regional pro-atrial natriuretic peptide (proANP) [7]. This peptide is related to the brain natriuretic peptide [8] and its prohormone form pro-brain natriuretic peptide (proBNP) [9–13], and both have been studied after cardiac surgery. ProANP has been described as a good marker for heart failure in adults [14,15], suggesting that it could be a good tool for stratifying risk before heart surgery.

Our hypothesis was that proADM and proANP levels prior to pediatric cardiac surgery might be higher in vulnerable patients that would need more intensive support throughout the post-operative period. The use of these biomarkers for this has not been described until now in children.

The main goal was to assess the usefulness of proADM and proANP before pediatric cardiac surgery in predicting a greater need for respiratory and inotropic support during the post-operative period. A secondary objective was to determine the relationship of these biomarkers with other complications that might arise after surgery.

## Materials and methods

This was a prospective observational study performed in the Pediatric ICU of a pediatric tertiary referral hospital. All patients admitted after pediatric cardiac surgery from 2012 to 2013 were included, independently of their need for cardiopulmonary bypass. We excluded patients

whose parents did not sign the informed consent form, as well as newborns admitted to the neonatal intensive care unit after cardiac surgery, because this is a different unit. The study was conducted in accordance with the Declaration of Helsinki and approved by the local research ethics committee (CEIm Fundación Sant Joan de Déu, Barcelona) and the institutional review board. Parental written informed consent was mandatory for recruited patients.

Serum levels of proADM and proANP were determined by immunofluorescence using Time-Resolved Amplified Cryptate Emission (TRACE) and a KRYPTOR analyzer (B·R·A·H·M·S Diagnostica GmbH, Henningsdorf, Germany). Detection limit values were 0.23nmol/L for proADM and 2.1pmol/L for proANP. A venous blood sample of 1 mL allowed for the analysis of both biomarkers. Values for proADM of <0.5 nmol/L and for proANP of <70 pmol/L were considered normal in children. The sampling times were always prior to cardiac surgery (pre-operative blood samples). The clinicians were blinded to the biomarker results during the study period.

Baseline data were collected: age at surgery, gender, weight, and previous surgeries. CHDs were divided into three categories following a classification based on the pathophysiology, as was done in previous studies [16]: (1) increased volume overload, (2) pressure overload involving the left ventricle or the right ventricle, (3) complex cyanotic CHD. Data about the heart surgery included the complexity of surgery score (STAT mortality scoring model [1]), type of surgery, and length of time for extracorporeal circulation, aortic cross clamping, and deep hypothermic circulatory arrest. The Pediatric Risk Mortality Score (PRISM III [17]) was calculated at admission. Respiratory support was considered as the total intubation time after surgery, measured by the number of hours on mechanical ventilation (MV) [18]. Hemodynamic support after cardiac surgery was analyzed using the vasoactive-inotropic score (VIS score [19]). The following complications after cardiac surgery were recorded: arrhythmia, low cardiac output syndrome (determined by echocardiography, left ventricular ejection fraction <40%), pulmonary hypertension with clinical repercussions, and the need for renal replacement therapies and/or mechanical circulatory support. A major complication was defined as any one of the following occurring within the first 30 days after surgery: mortality, renal failure, neurological deficit, arrhythmia requiring a pacemaker, mechanical circulatory support, paralyzed diaphragm, or an unplanned operation [20]. Poor outcome was defined as mortality, cardiac arrest, use of mechanical circulatory support, renal replacement therapy, or neurologic injury (stroke or seizure), as in previous studies [21]. Mortality was described as any death occurring during the stay in the ICU. Prolonged length of stay (LOS) was as a LOS in the 75th percentile.

As done by previous studies [22], in order to detect specific differences linked to age, patients were divided into 3 age groups: newborns (<1 month), infants (1–12 months), and children (>12 months–18 years).

The primary endpoints were a greater need for respiratory support, defined as the need for more than 72 hours of MV after cardiac surgery [23,24], and an increased need for inotropic support, assumed as a VIS score ≥20 points [21] during the first 24 hours after cardiac surgery. The secondary endpoints were the presence of complications after cardiac surgery (including poor outcome), prolonged LOS, and mortality.

The statistical analyses were performed using SPSS25.0®. The categorical variables were expressed as frequency and percentage, and the continuous variables were expressed as median and interquartile range (IQR). Data were analyzed via non-parametric tests, using the chi-squared test or Fisher's test as needed to compare categorical variables, and using the Mann-Whitney U test or the Kruskal-Wallis test for continuous variables. Spearman's rank correlation coefficient (r) was used to assess the associations between proANP and proADM before cardiac surgery and the continuous variables related to the patients' clinical characteristics, as well as the biomarkers' relationship with post-operative support needs. All tests were two-

tailed. A receiver operating characteristic (ROC) analysis was performed, analyzing biomarker cut-off points, area under the curve (AUC), sensitivity (Sn), specificity (Sp), positive predictive value (PPV), and negative predictive value (NPV). A multivariate logistic regression was used to assess the association between the predictors and the primary endpoints. We performed a logarithmic transformation of proANP (*Ln*-proANP) and proADM (*Ln*-proADM) values in the regression model because they have a non-parametric distribution. Variables incorporated into the multivariate model were those with a p-value of <0.2 in the univariate analysis and those with an elevated biological importance. All these results were expressed as odds ratio (OR) and 95% confidence interval (CI). A p-value of <0.05 was considered significant.

## Results

A total of 113 patients were analyzed. The flow chart of patients is shown in S1 Fig. Fifty-nine (50.9%) were males. The main clinical characteristics of the patients are included in Table 1.

Nine patients (7.8%) required prolonged respiratory support and three patients (2.7%) required increased inotropic support. Four patients (3.5%) presented major complications and two (1.8%) had poor outcomes. The median LOS was 3 days (IQR 2–5). Only one patient died in the ICU (0.9%) during the study, 40 days after the surgery; this was a patient who needed ECMO after surgery due to myocardial ischemia, with a subsequent evolution towards dilated cardiomyopathy.

**Table 1. Main demographic, surgical, and clinical characteristics of the patients.**

| Variables | Total patients (n = 113) |
|---|---|
| Male | 58 (51.3%) |
| Age (years) | 2.1 (0.6–6.6) |
| Newborns | 3 (2.7%) |
| Infants | 39 (34.5%) |
| Children | 71 (62.8%) |
| Weight (kg) | 12 (6.1–20) |
| Underlying disease | 25 (22.1%) |
| Previous cardiac surgery | 34 (30.1%) |
| CHD with increased volume overload | 59 (52.2%) |
| CHD with pressure overload involving the left ventricle or the right ventricle | 34 (30.1%) |
| Complex cyanotic CHD | 20 (17.7%) |
| STAT mortality category 1 | 36 (31.9%) |
| STAT mortality category 2 | 47 (41.6%) |
| STAT mortality category 3 | 25 (22.1%) |
| STAT mortality category 4 | 4 (3.5%) |
| STAT mortality category 5 | 1 (0.9%) |
| Univentricular physiology (Glenn or Fontan procedure) | 10 (8.8%) |
| Extracorporeal circulation required | 111 (98.2%) |
| Length of extracorporeal circulation (minutes) | 75 (55–100) |
| Length of aortic cross clamping (minutes) | 45 (29–68.5) |
| Deep hypothermic cardiac arrest required | 5 (4.4%) |
| Length of deep hypothermic cardiac arrest (minutes) | 25 (6.5–35) |
| PRISM III (points) | 3 (2–5) |
| Need for mechanical ventilation at admission | 49 (43.4%) |
| Intubation time after surgery (hours) | 4 (3–24) |

Continuous variables are expressed as median (interquartile range) and categorical variables are expressed as frequency (percentage).

## Biomarkers

There were no differences as regards gender: proADM was 0.51 nmol/L (IQR 0.36–0.65) for males and 0.42 nmol/L (IQR 0.33–0.65) for females, with p = 0.213, proANP was 123.5 pmol/L (IQR 78.9–225.7) vs. 105.6 pmol/L (IQR 69.9–257.6), with p = 0.539. No differences were detected as regards univentricular physiology, either: proADM was 0.46 nmol/L (IQR 0.31–0.59) vs. 0.47 nmol/L (IQR 0.35–0.65), with p = 0.479, and proANP, 99.2 pmol/L (IQR 68.2–133.9) vs. 117.6 pmol/L (IQR 73.3–249), with p = 0.169. Patients in a higher STAT mortality category presented higher values of both biomarkers (Fig 1). A negative correlation was detected between age and the biomarkers: r = -0.678 for proADM and r = -0.716 for proANP (both p<0.001). Fig 2 illustrates the different values of the biomarkers depending on the age group, showing higher values in the youngest patients, especially newborns, with statistically significant differences between groups. There were also statistically significant correlations between the biomarker values and the intubation time after surgery (r = 0.427, p<0.001 for proADM and r = 0.580, p<0.001 for proANP). All the correlations between biomarkers and the main continuous characteristics of the sample are included in Table 2.

## Primary endpoints

Patients with a greater need for respiratory and inotropic support after cardiac surgery presented higher values of both biomarkers in comparison with patients who did not. Fig 3 illustrates these results.

ProANP had a greater AUC than proADM for predicting increased respiratory support after cardiac surgery: AUC$_{proANP}$ 0.791 vs. AUC$_{proADM}$ 0.724, but there were no statistically significant differences between them (p = 0.494). Fig 4 represents the AUC of both biomarkers. The combination of the two biomarkers had an AUC of 0.796 (95% CI 0.626–0.966, p = 0.003). A value of proANP ≥325 pmol/L had a Sn of 66.7% (95% CI 29.9–92.5), Sp of 88.8% (95% CI 81.2–94.1), PPV of 33.3% (95% CI 13.3–59.0), and NPV of 96.9% (95% CI 91.3–99.4) for predicting increased respiratory support needs.

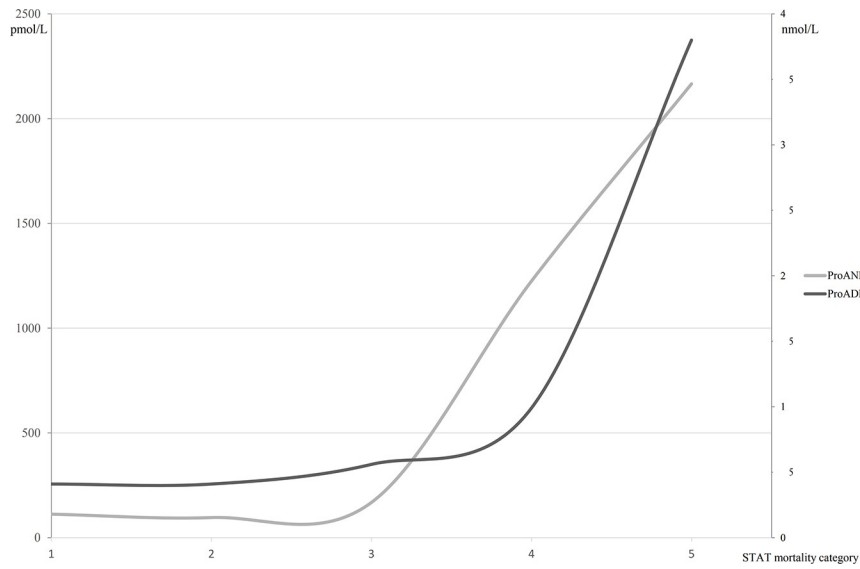

| STAT mortality category | ProADM (nmol/L) | ProANP (pmol/L) |
|---|---|---|
| 1 (n=36) | 0.41 (0.34-0.64) | 112.1 (66.8-238.8) |
| 2 (n=47) | 0.41 (0.29-0.59) | 96.3 (61.9-151.9) |
| 3 (n=25) | 0.56 (0.45-0.64) | 167.8 (100.6-334.3) |
| 4 (n=4) | 0.99 (0.59-0.64) | 1224.4 (199.5-3338.3) |
| 5 (n=1) | 3.8 | 2166 |
| p | 0.006 | 0.001 |

**Fig 1. Differences between pro-atrial natriuretic peptide (proANP, pmol/L) and pro-adrenomedullin (proADM, nmol/L) according to the mortality risk category (STAT mortality category) before cardiac surgery.** Values expressed as median (interquartile range) and compared using the Kruskal-Wallis test.

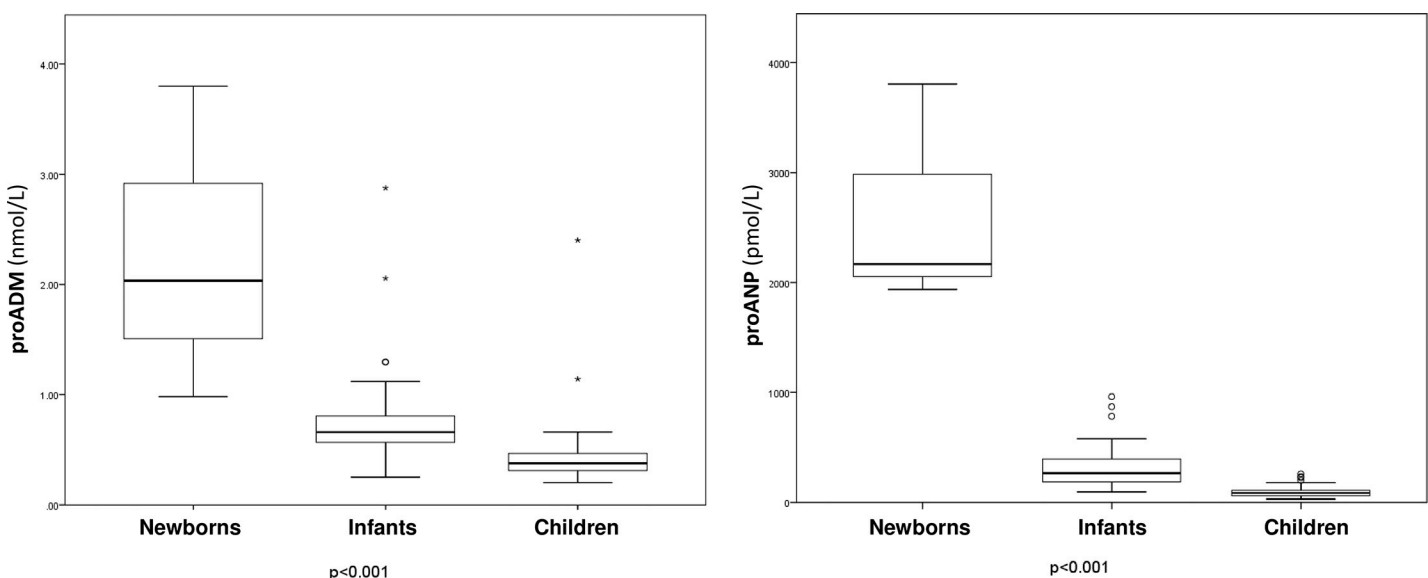

**Fig 2. Box plots showing the relationship between pro-atrial natriuretic peptide (proANP) and pro-adrenomedullin (proADM) and the three age groups.** Boxes show the interquartile range and the median. The comparison using the Kruskal-Wallis test yielded p<0.001 for the two biomarkers. Regarding the comparison between the two groups and proADM using the Mann Whitney U test: Newborns vs. Infants, p = 0.018; Newborns vs. Children, p = 0.005; Infants vs. Children, p<0.001. For proANP: Newborns vs. Infants, p = 0.004; Newborns vs. Children, p = 0.004; and Infants vs. Children, p<0.001.

ProANP also had a greater AUC for predicting an increased need for inotropic support: $AUC_{proANP}$ 0.948 (95% CI 0.874–1.000, p = 0.008) vs. $AUC_{proADM}$ 0.855 (95% CI 0.688–1.000, p = 0.037), without statistically significant differences between them (p = 0.137). Their use in combination had an AUC of 0.912 (95% CI 0.780–1.000, p = 0.015).

In the multivariate analysis, the variable that remained as the independent predictor for increased need for respiratory support was Ln-proANP, with an OR = 3.575 (95% CI 1.684–7.591, p = 0.001), Hosmer–Lemeshow goodness of fit test = 0.321, and Nagelkerke's $R^2$ = 0.260. The independent predictor for increased inotropic support needs was also Ln-proANP with an OR = 6.533 (95% CI 1.781–23.969, p = 0.005), Hosmer–Lemeshow goodness of fit test = 0.950, and Nagelkerke's $R^2$ = 0.451.

## Secondary endpoints

Patients who presented any kind of complication had higher values of proANP: 249 pmol/L (IQR 80.2–482.7) vs. 107.7 pmol/L (IQR 70.1–200.7), p = 0.025. No differences were detected

**Table 2. Correlations between biomarkers and clinical data, using Spearman's rank correlation coefficient (*r*).**

| Clinical data | ProADM *r* | *p* | ProANP *r* | *p* |
|---|---|---|---|---|
| Age (years) | -0.678 | <0.001 | -0.716 | <0.001 |
| PRISM III (points) | 0.318 | 0.001 | 0.300 | 0.001 |
| Extracorporeal circulation (minutes) | 0.196 | 0.039 | 0.328 | <0.001 |
| Aortic cross-clamping (minutes) | 0.253 | 0.011 | 0.382 | <0.001 |
| VIS score after 24h (points) | 0.100 | 0.290 | 0.218 | 0.020 |
| VIS score after 48h (points) | 0.289 | 0.002 | 0.321 | 0.001 |
| Intubation time after surgery (hours) | 0.427 | <0.001 | 0.580 | <0.001 |
| Length of stay (days) | 0.312 | 0.001 | 0.362 | <0.001 |

VIS: Vasoactive-inotropic score. ProADM: pro-adrenomedullin. ProANP: pro-atrial natriuretic peptide.

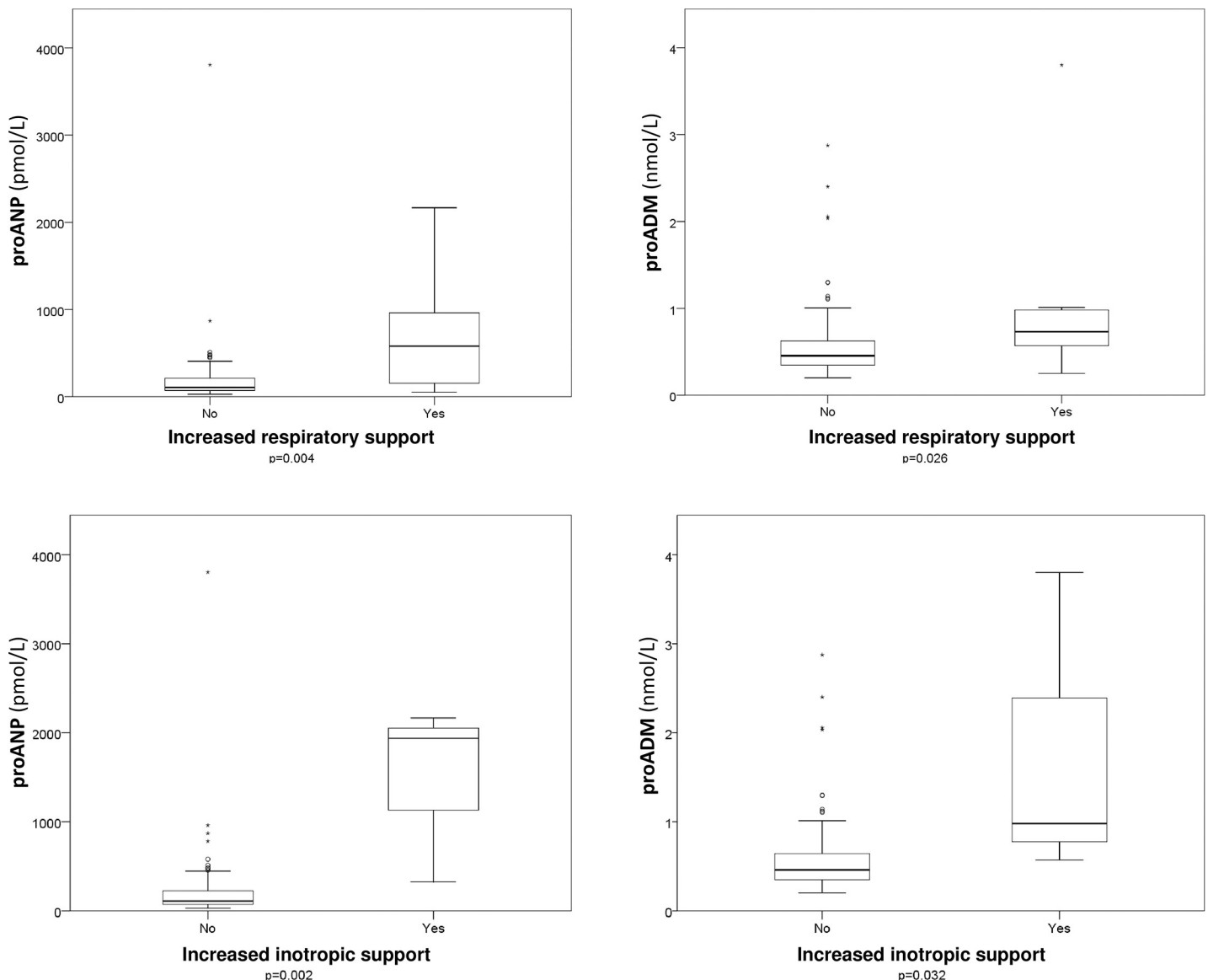

**Fig 3. Box plots showing the relationship between the biomarkers (proANP and proADM) and a greater need for respiratory or inotropic support after cardiac surgery.** Boxes show the interquartile range and the median. The comparison was done using the Mann-Whitney U test.

for proADM in the presence of any kind of complication (p = 0.295). Both biomarkers were higher in patients with poor outcome: proADM was 2.39 nmol/L (IQR 0.98–3.80) vs. 0.46 nmol/L (IQR 0.35–0.64) with p = 0.013, and proANP was 2052 pmol/L (IQR 1938–2166) vs. 112.0 pmol/L (IQR 73.5–226.4) with p = 0.001. Table 3 shows the biomarker values for the different types of complications.

## Discussion

As far as we know, this is the first study focused on the usefulness of measuring proANP and proADM before pediatric heart surgery. Our results suggest the ability of these biomarkers to predict an increased need for MV and inotropic support after cardiac surgery in children.

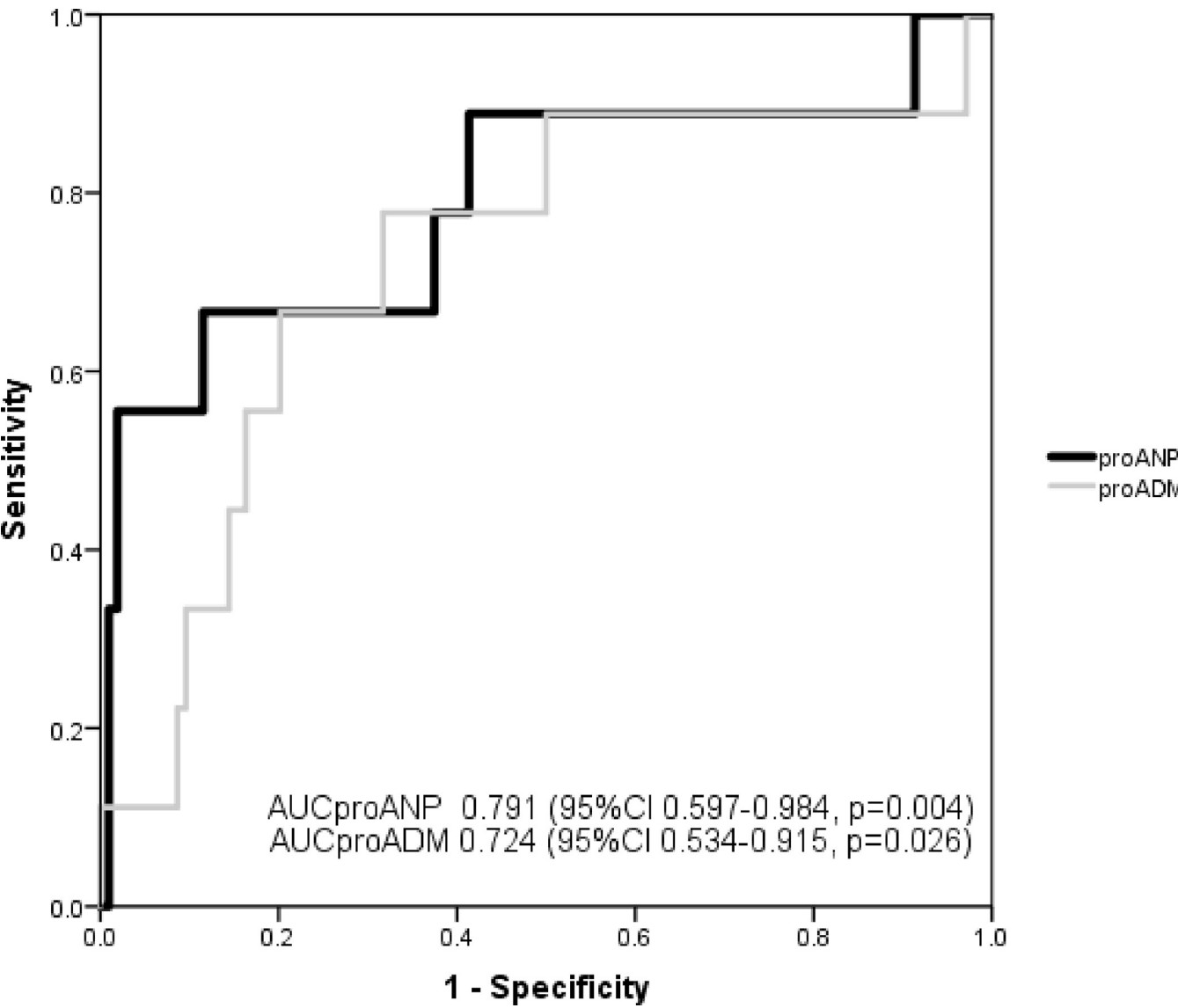

**Fig 4. Receiver operating characteristic (ROC) analysis of proANP and proADM with respect to predicting increased respiratory support needs in children after cardiac surgery.**

One of the advantages of using these biomarkers is that after drawing the blood sample, the result is available in only 30 minutes. Moreover, previous studies in adults have reported non-inferiority of proANP with respect to brain natriuretic peptide in many cardiovascular diseases [14,25]. For these reasons, at our center we decided to analyze them during the pre-surgery visit.

These two biomarkers have been particularly well-studied in sepsis and cardiovascular diseases in adults. High levels are associated with a poor prognosis in septic patients [26–30]. The role of proADM in sepsis is linked to its biological function as both a potent vasodilator and compensatory protein. In children, proADM determined within the first 24 hours following cardiac surgery seems to predict a greater need for postoperative respiratory support [31]. According to our results, proADM values are also useful before cardiac surgery, because high values can predict an increased need for respiratory support.

**Table 3. Values of proADM and proANP in the presence of complications after surgery.**

| Variables | | ProADM (nmol/L) | ProANP (pmol/L) |
|---|---|---|---|
| Pulmonary hypertension | Yes (n = 3) | 0.46 (0.44–0.51) | 80.2 (66.2-202-9) |
| | No (n = 110) | 0.47 (0.35–0.65) | 116.3 (73.7–229.3) |
| | p | 0.926 | 0.605 |
| Arrhythmia | Yes (n = 5) | 0.63 (0.42–1.53) | 343 (157.6–667.9) |
| | No (n = 108) | 0.46 (0.35–0.64) | 110.4 (73.4–222) |
| | p | 0.214 | 0.065 |
| Low cardiac output syndrome | Yes (n = 5) | 0.65 (0.46–0.66) | 482.7 (155.4–782.3) |
| | No (n = 108) | 0.46 (0.35–0.64) | 110.4 (73.5–226.4) |
| | p | 0.494 | 0.102 |
| Neurological disability | Yes (n = 2) | 2.39 (0.98–3.80) | 2052 (1938–2166) |
| | No (n = 111) | 0.46 (0.35–0.64) | 112 (73.5–226.4) |
| | p | 0.013 | 0.001 |
| Renal replacement therapy | Yes (n = 2) | 2.39 (0.98–3.80) | 2052 (1938–2166) |
| | No (n = 111) | 0.46 (0.35–0.64) | 112 (73.5–226.4) |
| | p | 0.013 | 0.001 |
| Extracorporeal circulatory support | Yes (n = 1) | 0.98 | 1938 |
| | No (n = 112) | 0.46 (0.35–0.64) | 113.7 (73.5–228.5) |
| | p | 0.230 | 0.053 |
| Major complication | Yes (n = 4) | 0.77 (0.34–3.09) | 1140 (135.3–2109) |
| | No (n = 109) | 0.46 (0.35–0.64) | 112 (73.5–226.4) |
| | p | 0.320 | 0.074 |
| Prolonged length of stay | Yes (n = 33) | 0.62 (0.47–0.75) | 191 (117.1–342.8) |
| | No (n = 80) | 0.42 (0.33–0.57) | 96.4 (66.7–177.0) |
| | p | 0.001 | 0.001 |
| Mortality in the pediatric intensive care unit | Yes (n = 1) | 0.98 | 1938 |
| | No (n = 112) | 0.46 (0.35–0.64) | 113.7 (73.5–228.5) |
| | p | 0.230 | 0.053 |

Values are expressed as median (IQR) and compared using the Mann-Whitney U test.

Extubation is an important stage after cardiac surgery due to the interaction between the lungs and heart that can be affected by MV. Not all patients are good candidates for early extubation (<6 hours after cardiac surgery). Today, anesthesiologists generally tend towards weaning in the operating theater if complications are not expected. The type of cardiac surgery and the patient's age are essential information when considering how soon to wean [23]. If a patient requires MV when they are admitted to the ICU, the weaning decision depends on the intensivist. We believe that proANP could be a good tool in this regard–a high value of proANP before cardiac surgery was related to an extended intubation time after surgery and could predict greater respiratory support needs after cardiac surgery, so this should alert the medical team not to proceed to weaning too soon. Many factors may influence the decision to extubate a child after cardiac surgery, so a single preoperative biomarker value would not serve as a stand-alone tool; rather, it could be another factor to consider before weaning.

ProANP rises in patients with heart failure due to its proprieties as a natriuretic, diuretic, and vasodilating agent. This protein is secreted in response to the increase in pressure inside the heart and in its dilating force. ProANP acts in the kidney, blood vessels, heart, adrenal glands, and adipocytes to carry out all its functions [32]. Its mechanism of action is linked to the compensatory strategy of the global system to regain balance after an injury. Its usefulness in adults

for monitoring cardiac failure and myocardial infarction has been demonstrated [33,34]. Some patients have heart failure before surgery despite the medical treatment, and these clinically unbalanced situations can lead to a worse post-operative evolution. This is consistent with our results, which showed that patients with higher pre-surgery proANP values required higher inotropic support after cardiac surgery. Furthermore, proADM also showed a good correlation with an increased need for inotropic support. This is consistent with the results of other studies, where proADM was higher in children with sepsis who required vasoactive drugs [35].

In addition, patients in a higher STAT mortality category presented higher levels of both biomarkers prior to surgery. To our knowledge, this has not been described before, so it might be useful information for the surgical team. Curiously enough, newborns also had higher values of the biomarkers. This may be due to the fact that this age range includes the most vulnerable patients and also has the greater number of complex surgeries. However, other biomarkers related to proANP, BNP and proBNP, also showed higher values in younger children, especially newborns [36,37]. This fact suggests that newborns may present higher baseline values of proANP in comparison with older children. However, the increase of proANP after birth is higher than that of proBNP, but proANP levels drop fast and after day 4 of life, these proANP levels normalize [38], and in our sample, the surgeries were performed after that age. ProANP and proADM have been suggested as markers for the severity of neonatal sepsis [39] and proANP was also proposed as a marker for the evolution of patent ductus arteriosus [40]. Moreover, the age variable was excluded from the multivariate analysis performed.

BNP and proBNP have been the most commonly studied biomarkers for heart failure and major complications after heart surgery in children [9,18,20,22,36,41–44]. However, their usefulness before cardiac surgery has not been standardized [13]. The release of proANP and proBNP and their functions run in parallel. BNP is also secreted by myocytes as a result of tension in the wall of the heart, and its levels seem to depend on the type of CHD rather than its severity [9–12,22]. Both natriuretic peptides are good biomarkers of left ventricular impairment in adults with heart failure [45] and their correlation is optimal [15]. More has been published about BNP and proBNP in children due to its easier measurement and better reproducibility when compared to ANP [8]. However, the development of immunoassays [7] that analyze proANP has reduced the gap between the advantages of each, and our results regarding proANP are promising.

After seeing these results, one question comes to mind: which biomarker is better before cardiac surgery, proADM or proANP? Statistically speaking, both of them have demonstrated the ability to identify the most vulnerable patients before cardiac surgery. However, looking at the absolute numbers and the multivariate analysis, we believe that proANP is more useful and easier to interpret, because the differences of values between children with more or less need for inotropic and respiratory support are remarkably higher.

This is a preliminary study that analyzes, for the first time, the ability of these two biomarkers before pediatric heart surgery to predict the risk of complications after surgery. The most important limitations are the single-center design and the sample size. More studies are needed to increase our knowledge about these biomarkers, especially those focused on their usefulness before cardiac surgery and their relationship with the type of CHD and the possible individual variations. Younger patients should be analyzed separately, especially newborns, who are the most vulnerable patients, in order to explore cut-off points for each age group. The comparison of proANP and proBNP before cardiac surgery could also be of interest, since proBNP has been considered as the only biomarker for heart failure after cardiac surgery in children until now [36]. Analyzing proANP before cardiac surgery, combined with looking at a mortality risk score like the STAT mortality category, could change the stratification of these patients due to the ability of this biomarker to identify the most vulnerable patients.

In conclusion, a high proANP value before cardiac surgery should alert anesthesiologists and intensivists that the patient will probably need longer MV and more inotropic support during the post-operative period. Nevertheless, more studies are needed to confirm these results and explore further applications for these biomarkers.

## Supporting information

**S1 Fig. Flow chart of patients.**
(TIFF)

## Author Contributions

**Conceptualization:** Sara Bobillo-Perez, Monica Girona-Alarcon, Iolanda Jordan.

**Formal analysis:** Sara Bobillo-Perez, Monica Girona-Alarcon, Iolanda Jordan.

**Investigation:** Sara Bobillo-Perez, Monica Girona-Alarcon, Iolanda Jordan.

**Validation:** Sara Bobillo-Perez, Monica Girona-Alarcon, Iolanda Jordan.

**Visualization:** Patricia Corniero, Anna Sole-Ribalta, Monica Balaguer, Elisabeth Esteban, Anna Valls, Francisco Jose Cambra.

**Writing – original draft:** Sara Bobillo-Perez, Monica Girona-Alarcon, Iolanda Jordan.

**Writing – review & editing:** Patricia Corniero, Anna Sole-Ribalta, Monica Balaguer, Elisabeth Esteban, Anna Valls, Francisco Jose Cambra.

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
