## [Decision Letter · Decision Letter 0]

20 May 2020

PONE-D-20-12352

Pro-atrial natriuretic peptide and pro-adrenomedullin before cardiac surgery in children. Can we predict the future?

PLOS ONE

Dear Dr. Jordan,

Thank you for submitting your manuscript to PLOS ONE. After careful consideration, we feel that it has merit but does not fully meet PLOS ONE’s publication criteria as it currently stands. Therefore, we invite you to submit a revised version of the manuscript that addresses the points raised during the review process.

The manuscript has been carefully evaluated by 2 external reviewers and they found the manuscript potentially of interest. However, the referees have identified some conceptual and methodological problems and they have required additional information and clarifications from the authors that need to be provided. 

We would appreciate receiving your revised manuscript by Jul 04 2020 11:59PM. To enhance the reproducibility of your results, we recommend that if applicable you deposit your laboratory protocols in protocols.io, where a protocol can be assigned its own identifier (DOI) such that it can be cited independently in the future. For instructions see: http://journals.plos.org/plosone/s/submission-guidelines#loc-laboratory-protocols

We look forward to receiving your revised manuscript.

Kind regards,

Claudio Passino, MD

Academic Editor

PLOS ONE

Journal Requirements:

"The study was conducted in accordance with the Declaration of Helsinki and approved by the local ethical investigational committee and the institutional review board. Parental written informed consent was mandatory for recruited patients."

Reviewers' comments:

Reviewer's Responses to Questions

**Comments to the Author**

1. Is the manuscript technically sound, and do the data support the conclusions?

Reviewer #1: Yes

Reviewer #2: Yes

2. Has the statistical analysis been performed appropriately and rigorously? 

Reviewer #1: Yes

Reviewer #2: Yes

3. Have the authors made all data underlying the findings in their manuscript fully available?

Reviewer #1: Yes

Reviewer #2: No

4. Is the manuscript presented in an intelligible fashion and written in standard English?

Reviewer #1: Yes

Reviewer #2: Yes

5. Review Comments to the Author

Reviewer #1: I read with interest the article entitled “ Pro-atrial natriuretic peptide and pro-adrenomedullin before cardiac surgery in children. Can we predict the future?” The subject is interesting and quite innovative, sample size good. Results are interesting and consisting. The article however has a few limitations that should be addressed.

Major comments

-English and format need to be revised.

-Abstract and introduction are a bit wordy.

Methods:

- A prospective study conducted almost ten years ago (2012-2013)?

- Sample time is only pre-operative? When? The lack of post-surgical values is a clear limitation of the present work

-Complicated outcome and major complications after pediatric cardiac surgery definitions should be consistent with literature. We advise to see and cite similar articles on other biomarkers

Cantinotti M, Giordano R, Scalese M, Molinaro S, Della Pina F, Storti S, Arcieri L, Murzi B, Marotta M, Pak V, Poli V, Iervasi G, Kutty S, Clerico A.Prognostic role of BNP in children undergoing surgery for congenital heart disease: analysis of prediction models incorporating standard risk factors. Clin Chem Lab Med. 2015 Oct;53(11):1839-46.

Cantinotti M, Storti S, Lorenzoni V, Arcieri L, Moschetti R, Murzi B, Spadoni I, Passino C, Clerico A.

The combined use of neutrophil gelatinase-associated lipocalin and brain natriuretic peptide improves risk stratification in pediatric cardiac surgery.

Clin Chem Lab Med. 2012 Nov;50(11):2009-17

Cantinotti M, Lorenzoni V, Storti S, Moschetti R, Murzi B, Marotta M, Crocetti M, Molinaro S, Clerico A, Portman M, Iervasi G Thyroid and brain natriuretic Peptide response in children undergoing cardiac surgery for congenital heart disease- age-related variations and prognostic value. Circ J. 2013;77(1):188-97.

Methods and results:

-Did you evaluate differences among neonates, infants, and children? If not, it may be of worth since for BNP the trend was quite different. You state something in the limitation section, but this point should be elucidated and detailed better, also in the tables.

-In the tables please specify the type of CHDs

-Inclusion/exclusion criteria re not well defined. Did you include only cardiopulmonary bypass surgery?

Discussion: comparison with other biomarkers are required

Limitation section need to implement.

Minor comments

CVC= I would rather use pediatric cardiac surgery without abbreviation

Reviewer #2: To the Authors

General Considerations

The aim of this study was to evaluate the utility of proANP and proADM determined before cardiovascular surgery for predicting need for high respiratory or inotropic support in the post-operative period. Authors enrolled 113 patients (51% males, median age 2.1 years, interquartile range 0.6-6.6 years) admitted to the neonatal intensive care unit after cardiovascular surgery. Authors considered as primary endpoints were a high respiratory support, considered as the need for more than 72 hours of mechanical ventilation after cardiovascular surgery, and a high inotropic support, in the first 24 hours after surgery. Furthermore, Secondary endpoints were the presence of complications after surgery, including Prolonged length of stay in ICU and mortality. The most important result of this study is that a high proANP determined before cardiovascular surgery may identify those patients who will need higher respiratory and inotropic support in the post-operative period.

The manuscript is concise and the results are clearly reported. I have only one observation to address to the Authors. It is well known that B-type natriuretic peptides (especially BNP and NT-proBNP) are the first line biomarkers for screening of heart failure and are also reliable prognostic markers in children undergoing cardiac surgery (Authors should cite the important review: Cantinotti M et al. Heart Fail Rev 2014;19:727-42). Therefore, the original data related to this article are that also some A-type natriuretic peptides, such as proANP, can have a prognostic role in paediatric patients admitted to the neonatal intensive care. Authors should explain because clinicians should prefer the assay of proANP in respect to that of BNP or NT-proBNP in children undergoing cardiac surgery. Accordingly, an important limitation of this study is that Authors should due to compare the prognostic accuracy of proANP to that of BNP or NT-proBNP. Authors should discuss this important point in revised manuscript.

6. PLOS authors have the option to publish the peer review history of their article (what does this mean?). If published, this will include your full peer review and any attached files.

Reviewer #1: Yes: Massimiliano Cantinotti

Reviewer #2: No

---

## [Author Response · Author response to Decision Letter 0]

29 Jun 2020

Reviewer #1: I read with interest the article entitled “ Pro-atrial natriuretic peptide and pro-adrenomedullin before cardiac surgery in children. Can we predict the future?” 

The subject is interesting and quite innovative, sample size good. Results are interesting and consisting. The article however has a few limitations that should be addressed.

Dear Reviewer, Thank you very much for your valuable comments. Undoubtedly, all your suggestions have helped improve the quality of this manuscript. We hope we have adequately answered all queries. Please let us know if further changes are required.

Major comments

-English and format need to be revised.

->Dear reviewer, the manuscript was revised by a professional English translator. The changes related to the language editing were not marked. The format follows the journal guidelines.

-Abstract and introduction are a bit wordy.

->Dear reviewer, following your recommendations, changes have been made in the abstract and introduction. 

Methods:

-A prospective study conducted almost ten years ago (2012-2013)? Sample time is only pre-operative? When? The lack of post-surgical values is a clear limitation of the present work.

-> Dear reviewer, the project of the present prospective study included the analysis of proADM and proANP before surgery, immediately after surgery and between the 24–36 hours after surgery. However, a primary analysis was made only with the samples in the postoperative period (published in 2019, reference 31), and the pre-surgery data were excluded, loosing this information. Last year, while we were planning a new study about the use of biomarkers before surgery to predict risk in this kind of patients, we recovered this data to perform an exploratory analysis. Surprisingly, the results were interesting and reviewing the articles in last years, we felt that this data needed to be shared with others. No publication has explored the role of proANP or proADM in these patients until now, and we decided to work on this manuscript, focused exclusively on biomarkers before surgery to predict risk after surgery. 

The blood sample was obtained in the Anaesthesia evaluation before cardiac surgery.

-Complicated outcome and major complications after pediatric cardiac surgery definitions should be consistent with literature. We advise to see and cite similar articles on other biomarkers.

->Dear Reviewer, thank you for your suggestion. We have added the definition of major complications according to the reference Cantinotti M, et al. Clin Chem Lab Med. 2015 Oct;53(11):1839-46 . The definition of poor outcome is based on previous studies (Gaies MG, et al. Pediatr Crit Care Med. 2014;15: 529–537), and we have decided to keep it in the article because our data revealed statistically significant differences regarding biomarkers and poor outcomes. In contrast, no differences were detected regarding biomarkers and major complications. However, a new analysis was made, considering major complications, and the new results have been included in table 3.

We also added the evaluation of respiratory support as intubation time after surgery, adding the reference suggested: Cantinotti M, et al. Clin Chem Lab Med. 2012 Nov;50(11):2009-17

Methods and results:

-Did you evaluate differences among neonates, infants, and children? If not, it may be of worth since for BNP the trend was quite different. You state something in the limitation section, but this point should be elucidated and detailed better, also in the tables.

->Dear reviewer, thank you for your suggestion. According to it, we evaluated the age-related differences. The division in 3 age groups was made based on the previous literature (Cantinotti M. Circ J. 2013;77(1):188-97. Line 123-124. The two biomarkers were analysed regarding the age-group, and differences were observed (new figure 2), such as the correlation biomarkers-age suggested. However the multivariate did not show differences, and 2 of the 3 newborns also presented major complications and high support in the PICU. We added information regarding age and biomarkers to the discussion section. A study that analysed proANP and proBNP showed that proANP presented a higher increased in the two first days of life and later presented a drop until normalize its levels since day 4th of life. In contrast, proBNP presented a lower increase but maintained high levels for more days after life (Mir TS, et al. Plasma concentrations of aminoterminal pro atrial natriuretic peptide and aminoterminal pro brain natriuretic peptide in healthy neonates: Marked and rapid increase after birth. Pediatrics. 2003;112:896–899). Due to the differences between patients, we consider that more studies are needed to analyse separately newborns for being a special population, as we stated in the limitation subsection. 

-In the tables please specify the type of CHDs

->Dear Reviewer, considering your suggestion, we have classified the patients in 3 categories according to a patho-physiological classification based on previous articles related to biomarkers (Especially Cantinotti et al.(2011) Review. Clin Chem Lab Med 49:567–580). Please, let us know if you consider that other classification is more suitable, such as the one based on the preoperative echocardiogram (Clancy, et al. J ThoracCardiovascSurg, 2000). 

Class I: Two ventricle heart without arch obstruction 88 (77.9%)

Class II: Two ventricle heart with arch obstruction 8 (7.1%)

Class III: Single ventricle heart without arch obstruction 8 (7.1%)

Class IV: Single ventricle heart with arch obstruction 9 (8%)

-Inclusion/exclusion criteria are not well defined. Did you include only cardiopulmonary bypass surgery?

->Dear Reviewer, we included all the cardiac surgeries. Cardiopulmonary bypass was not required in two patients (2 Glenn procedures). We added this information to the table 1, and we also modified the explanation of the inclusion criteria.

Discussion: comparison with other biomarkers are required

->Dear Reviewer, we have added the comparison with the main prognostic biomarkers in cardiac surgery in children, the BNP and proBNP. Due to the length of the manuscript, the comparison is brief. Please, let us know if you consider that more information is required. 

Limitation section need to implement.

->Dear Reviewer, the limitation subsection has been modified, especially the lack of comparison of proANP and proBNP. Our results confirm the usefulness of proANP in these patients, as you demonstrated regarding BNP and proBNP. This is the first time this analysis is performed. Afterwards, the comparison of these natriuretic peptides would be interesting. We chose proANP to explore new biomarkers and also to analyse the possibility to combine proANP and proADM to enhance our power of prediction, although our data did not confirm this.

Minor comments

CVC= I would rather use pediatric cardiac surgery without abbreviation

->Dear Reviewer, the abbreviation CVS was removed following your suggestion. 

Reviewer #2:

 To the Authors

General Considerations

The aim of this study was to evaluate the utility of proANP and proADM determined before cardiovascular surgery for predicting need for high respiratory or inotropic support in the post-operative period. Authors enrolled 113 patients (51% males, median age 2.1 years, interquartile range 0.6-6.6 years) admitted to the neonatal intensive care unit after cardiovascular surgery. Authors considered as primary endpoints were a high respiratory support, considered as the need for more than 72 hours of mechanical ventilation after cardiovascular surgery, and a high inotropic support, in the first 24 hours after surgery. Furthermore, Secondary endpoints were the presence of complications after surgery, including Prolonged length of stay in ICU and mortality. The most important result of this study is that a high proANP determined before cardiovascular surgery may identify those patients who will need higher respiratory and inotropic support in the post-operative period.

The manuscript is concise and the results are clearly reported. 

I have only one observation to address to the Authors. It is well known that B-type natriuretic peptides (especially BNP and NT-proBNP) are the first line biomarkers for screening of heart failure and are also reliable prognostic markers in children undergoing cardiac surgery (Authors should cite the important review: Cantinotti M et al. Heart Fail Rev 2014;19:727-42). Therefore, the original data related to this article are that also some A-type natriuretic peptides, such as proANP, can have a prognostic role in paediatric patients admitted to the neonatal intensive care. Authors should explain because clinicians should prefer the assay of proANP in respect to that of BNP or NT-proBNP in children undergoing cardiac surgery. Accordingly, an important limitation of this study is that Authors should due to compare the prognostic accuracy of proANP to that of BNP or NT-proBNP. Authors should discuss this important point in revised manuscript.

->Dear Reviewer, thank you very much for your comments. There are many studies focused on the usefulness of BNP and NT-proBNP in children undergoing cardiac surgery. We planned to explore new biomarkers different to BNP and NT-proBNP. The present study analysed two biomarkers that nobody had evaluated before paediatric cardiac surgery until now. As we said in the Discussion section: “One of the advantages of using these biomarkers is that after drawing the blood sample, the result is available in only 30 minutes. Moreover, previous studies in adults have reported non-inferiority of proANP with respect to brain natriuretic peptide in many cardiovascular diseases [14,24]. Because of these reasons, in our centre we decided to analyse them during the pre-surgery visit.” Data regarding the comparison of proANP and proBNP have been added to the Discussion subsection. 

As we stated in the limitations subsection, it is needed a comparison of proANP and NT-proBNP in these patients. However, we considered that these results are relevant as it is the first time that these biomarkers have been evaluated. Please, let us know if more information is required.

---

## [Decision Letter · Decision Letter 1]

7 Jul 2020

Pro-atrial natriuretic peptide and pro-adrenomedullin before cardiac surgery in children. Can we predict the future?

PONE-D-20-12352R1

Dear Dr. Jordan,

We’re pleased to inform you that your manuscript has been judged scientifically suitable for publication and will be formally accepted for publication once it meets all outstanding technical requirements.

Kind regards,

Claudio Passino, MD

Academic Editor

PLOS ONE

Additional Editor Comments (optional):

Reviewers' comments:

Reviewer's Responses to Questions

**Comments to the Author**

1. If the authors have adequately addressed your comments raised in a previous round of review and you feel that this manuscript is now acceptable for publication, you may indicate that here to bypass the “Comments to the Author” section, enter your conflict of interest statement in the “Confidential to Editor” section, and submit your "Accept" recommendation.

Reviewer #1: All comments have been addressed

Reviewer #2: All comments have been addressed

2. Is the manuscript technically sound, and do the data support the conclusions?

Reviewer #1: Yes

Reviewer #2: Yes

3. Has the statistical analysis been performed appropriately and rigorously? 

Reviewer #1: Yes

Reviewer #2: Yes

4. Have the authors made all data underlying the findings in their manuscript fully available?

Reviewer #1: Yes

Reviewer #2: Yes

5. Is the manuscript presented in an intelligible fashion and written in standard English?

Reviewer #1: Yes

Reviewer #2: Yes

6. Review Comments to the Author

Reviewer #1: Authors correctly addressed all points raised by the reviewers.

I have just a request. If data have been partly already published, this should be clearly stated. Again, quite strange a prospective study date bake 2013

Reviewer #2: Authors revised the manuscript in accordance with the suggestions made by the Reviewers. The manuscript is now significantly improved.

7. PLOS authors have the option to publish the peer review history of their article (what does this mean?). If published, this will include your full peer review and any attached files.

Reviewer #1: **Yes: **Massimiliano Cantinotti

Reviewer #2: No

---

## [Editor Report · Acceptance letter]

10 Jul 2020

PONE-D-20-12352R1 

Pro-atrial natriuretic peptide and pro-adrenomedullin before cardiac surgery in children. Can we predict the future? 

Dear Dr. Jordan:

I'm pleased to inform you that your manuscript has been deemed suitable for publication in PLOS ONE. Congratulations! Your manuscript is now with our production department. 

Kind regards, 

on behalf of

Prof. Claudio Passino 

Academic Editor

PLOS ONE